# Human pyramidal to interneuron synapses are mediated by multi-vesicular release and multiple docked vesicles

Gábor Molnár[1][†], Márton Rózsa[1][†], Judith Baka[1], Noémi Holderith[2], Pál Barzó[3], Zoltan Nusser[2], Gábor Tamás[1]*

[1]MTA-SZTE Research Group for Cortical Microcircuits, Department of Physiology, Anatomy and Neuroscience, University of Szeged, Szeged, Hungary; [2]Institute of Experimental Medicine, Hungarian Academy of Sciences, Budapest, Hungary; [3]Department of Neurosurgery, University of Szeged, Szeged, Hungary

**Abstract** Classic theories link cognitive abilities to synaptic properties and human-specific biophysical features of synapses might contribute to the unparalleled performance of the human cerebral cortex. Paired recordings and multiple probability fluctuation analysis revealed similar quantal sizes, but 4-times more functional release sites in human pyramidal cell to fast-spiking interneuron connections compared to rats. These connections were mediated on average by three synaptic contacts in both species. Each presynaptic active zone (AZ) contains 6.2 release sites in human, but only 1.6 in rats. Electron microscopy (EM) and EM tomography showed that an AZ harbors 4 docked vesicles in human, but only a single one in rats. Consequently, a Katz's functional release site occupies ~0.012 $\mu m^2$ in the human presynaptic AZ and ~0.025 $\mu m^2$ in the rat. Our results reveal a robust difference in the biophysical properties of a well-defined synaptic connection of the cortical microcircuit of human and rodents.

*For correspondence: gtamas@bio.u-szeged.hu

[†]These authors contributed equally to this work

**Competing interests:** The authors declare that no competing interests exist.

## Introduction

The human cerebral cortex is considered to be one of the most complex biological structures, endowing us to perform sophisticated cognitive tasks. The layered architecture, cellular elements and synaptic connectivity of the human neocortex have been investigated and compared to those of other mammals such as primates, carnivores, and rodents (*Komlosi et al., 2012*; *Mohan et al., 2015*; *Molnar et al., 2008*; *Olah et al., 2007*; *Prince and Wong, 1981*; *Schwartzkroin and Knowles, 1984*; *Verhoog et al., 2013*; *Wang et al., 2015*; *Yanez et al., 2005*). Although, there are species-specific differences in the layering, long-range connections and intrinsic electrical and morphological properties of nerve cells (*Mohan et al., 2015*; *Wang et al., 2015*), the cellular elements and local synaptic circuits were found to be rather stereotyped. For example, similar types of cortical GABAergic interneurons (INs), pyramidal cells (PCs) and comparable local synaptic connectivity patterns were reported in rodent and human neocortices (*Molnar et al., 2008*; *Olah et al., 2007*; *Yanez et al., 2005*). However, classic theories suggest that synaptic properties also contribute to cognitive abilities (*Bliss and Lomo, 1973*; *Buzsaki, 2015*; *Hebb, 1949*; *Singer, 1995*) and recent studies revealed differences in synaptic communication between cell types that are highly conserved among mammals. Human cortical PC to PC connections display spike timing-dependent long-term plasticity similar to that found in rodents, but with altered plasticity-inducing activity patterns (*Verhoog et al., 2013*). It has also been demonstrated that the outputs of human PCs can be so powerful that individual presynaptic action potentials result in suprathreshold postsynaptic responses in GABAergic neurons and polysynaptic series of events downstream in the network

(*Komlosi et al., 2012*; *Molnar et al., 2008*). These powerful PC to IN connections are not characteristic to nonhuman tissue and were speculated to be important in reliable propagation of information in the human neocortex (*Woodruff and Yuste, 2008*). Here we aimed at determining set out to determine whether underlying biophysical properties and structural features of these connections are different in our species compared to one of the most widely used rodent model organisms, the rat.

## Results

### Quantal parameters of pyramidal cell to fast spiking interneuron EPSCs in the human and rat cerebral cortex

We compared the functional properties of synaptic connections made by PCs onto fast-spiking INs in the layer 2/3 of human (n = 39) and rats (n = 26; *Figure 1*) neocortex. Acute brain slices were prepared from small blocks of non-pathological samples of human cortical tissue resected in surgery from female (n = 15, aged 53 ± 13 years) and male (n = 11, aged 43 ± 24 years) patients (*Komlosi et al., 2012*; *Molnar et al., 2008*), from somatosensory cortex of juvenile (P18 - 28, n = 9) and prefrontal cortex of adult (P53 - 65, n = 17) male rats using standard acute slice preparation procedures (*Olah et al., 2007*). Both presynaptic PCs and postsynaptic INs were whole-cell recorded with biocytin-containing intracellular solutions, allowing *post hoc* identification of the recorded cell types and the number of synaptic contacts between them. Neurons were chosen based on their typical membrane and firing properties. Postsynaptic INs were identified as either axo-axonic (AACs, human: n = 6) or basket (BCs, human: n = 18; rat: n = 8) cells based on their characteristic axonal cartridges or axonal branches forming perisomatic baskets, respectively (*Klausberger and Somogyi, 2008*; *Figure 1B and F*). Reconstructions allowed the measurement of the length of dendritic (BCs, n = 9, 2912 ± 1076 µm; AACs, n = 5, 2897 ± 1546 µm, p = 1, MW U-test) and axonal (BCs, n = 7, 27,489 ± 14,080 µm; AACs, n = 5, 20732 ± 3332 µm, p = 0.41) arbors within our human slices. Excitatory postsynaptic currents (EPSCs), recorded from INs in response to single presynaptic action potentials showed a wide range of amplitudes in both species (human: 9.0–1477.1 pA; rat: 7.7–236.3 pA), with a significantly larger mean amplitude in human INs (human: 258.8 ± 272.8 pA; rat: 75.8 ± 58.7 pA; p<0.001, MW U-test, *Figure 1C and G*). Unitary EPSCs in human and rats had similar latencies (human: 1.04 ± 0.4 ms, n= 37; rat: 1.1 ± 0.38 ms, n = 25; p = 0.79, MW U-test), rise (10–90%, human: 0.44 ± 0.16 ms, n = 39; rat: 0.43 ± 0.14 ms; n = 26; p = 0.95, MW U-test) and decay (37%, human: 1.46 ± 0.83 ms, n = 38; rat: 1.7 ± 0.8 ms, n = 26; P = 0.18, MW U-test) times and the larger amplitude with similar decay resulted in a significantly larger EPSC charge in human (human: 553.6 ± 593.3 fC, n=38; rat: 123.1 ± 107.6 fC, n = 26; p<0.001, MW U-test; *Figure 1I*).

To determine quantal parameters of the unitary EPSCs, we performed multiple probability fluctuation analysis (MPFA, *Clements and Silver, 2000*; *Silver, 2003*; human: n = 10, rat: n = 12; *Figure 2*). Different release probability conditions were imposed by altering $[Ca^{2+}]_o$ and $[Mg^{2+}]_o$ in the extracellular solution during recordings (*Silver, 2003*) and measured the means and variances of EPSC charge (*Figure 2B and D*). These recordings were performed in the presence of an NMDA and a cannabinoid receptor antagonist (20 µM D-AP-5 and 10 µM AM251, respectively) to prevent NMDA channel openings, which might reduce variances and induce long-term plasticity (*Silver, 2003*; *Saviane and Silver, 2007*) and to exclude undesirable retrograde short- or long-term modification of glutamatergic transmission (*Lee et al. 2010*, *Péterfi et al. 2012*) during MPFA. The low affinity competitive non-NMDA receptor antagonist γDGG (0.5 mM) was included to prevent AMPA receptor saturation (*Sakaba et al., 2002*; *Scheuss et al., 2002*; *Wadiche and Jahr, 2001*).

To avoid rundown of EPSCs during long-lasting recordings, 10 mM L-glutamate was added to the internal solution of the presynaptic cell (*Biro et al., 2005*; *Ishikawa et al., 2002*). As expected, the amplitudes of human and rat unitary EPSCs were effectively modulated by altering $[Ca^{2+}]_o$ and $[Mg^{2+}]_o$ (*Figure 2A and C*), consistent with changes in the probability of release ($P_r$) of a vesicle at a functional release site. EPSC failure rates were significantly lower in human neurons (p<0.012, MW U-test; *Figure 2B and D*) in each condition compared to rats ($[Ca^{2+}]_o$: 1 mM: 15.8 ± 18.7% vs. 62.8 ± 34.2%, 1.5 mM: 2.0 ± 4.0% vs. 39.1 ± 24.7%, 2 mM: 0 vs. 26.6 ± 21.7%, 4 mM: 0 vs. 8.4 ± 10.2%), indicating either different vesicle $P_r$ or number of functional release sites ($N_{frs}$). This approach resulted in a large, 10- and 5.5-fold increase in the mean charge transferred by human and rat

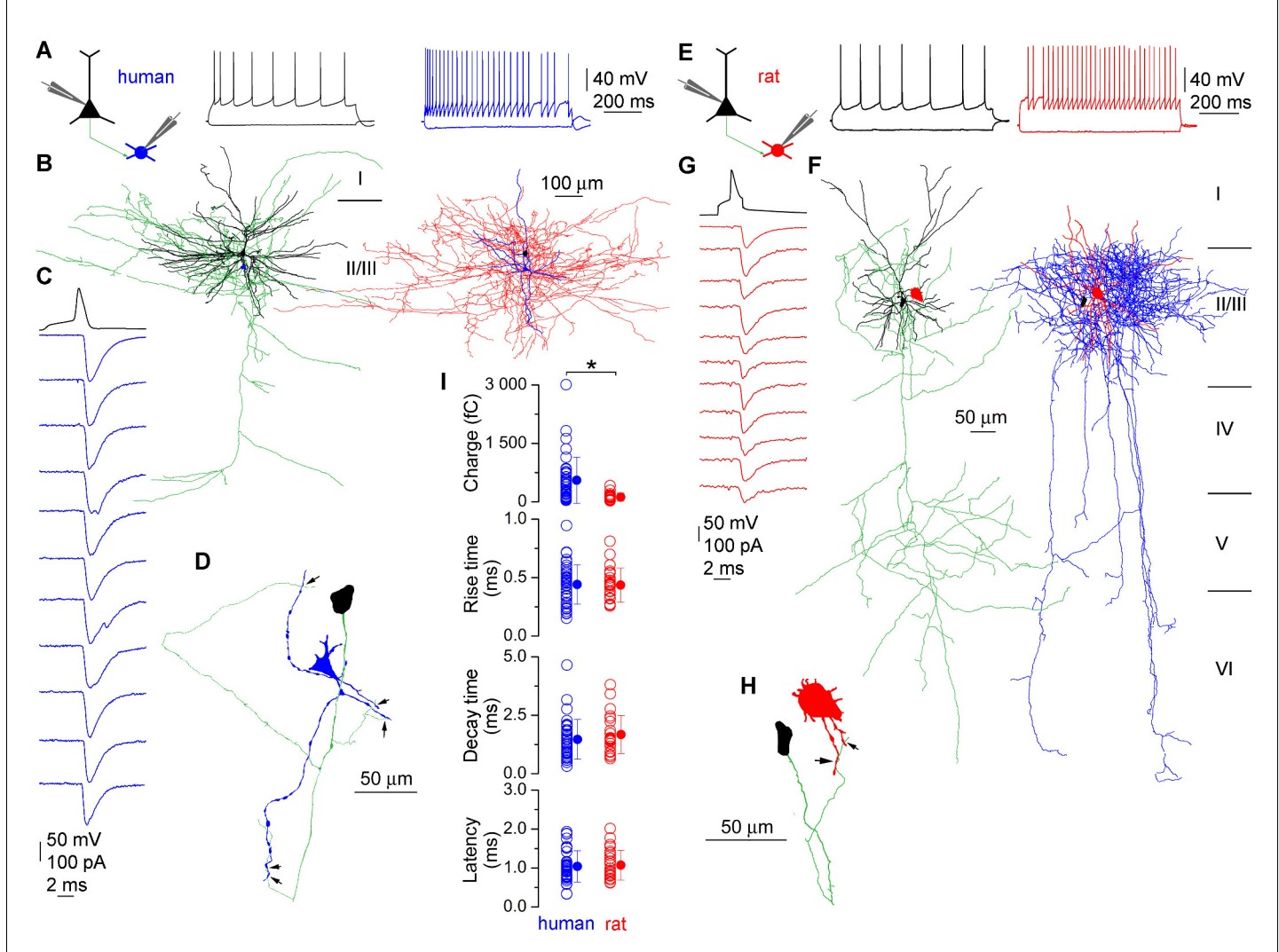

**Figure 1.** Monosynaptic excitatory connections from pyramidal cells to interneurons in the human and rat cerebral cortex. (A) Firing pattern of a presynaptic pyramidal cell (black) and a postsynaptic basket cell (blue) simultaneously recorded in an acute slice of the human cerebral cortex. (B) LM reconstruction of the recorded pyramidal cell (left, soma and dendrites: black, axons: green) and basket cell (right, soma and dendrites: blue, axons: red). Scale bars apply to both cells. Reconstructions of the cells are separated for clarity with relative positions of the somata (pyramidal cell: black, basket cell: blue) indicated on both panels. Roman numbers represent cortical layers. (C) Presynaptic action potentials of the pyramidal cell (black) elicited unitary EPSCs in the basket cell held at −70 mV (individual traces: blue, average: top blue trace). (D) Route of the presynaptic axon (green) from the PC soma (black) to putative synaptic contacts (arrows) on the basket cell dendrites (blue). (E–H) A rat pyramidal cell to basket cell connection is presented similar to panels A–D. (I) Human and rat unitary EPSCs have similar latencies, rise and decay times, but the human EPSC charge is significantly larger. Data presented as mean ± SD.

The following figure supplement is available for figure 1:

**Figure supplement 1.** Human EPSC parameters according to age and gender.

unitary EPSCs, respectively from the lowest to the highest $[Ca^{2+}]_o$ (*Figure 2E*), which is a prerequisite for reliable determination of the quantal parameters (*Silver, 2003*). By fitting the mean vs. variance relationship with a parabola, we obtained an estimate of quantal size (*q*), $P_r$ and $N_{frs}$ for each cell pair. Statistical comparisons of data from all cells revealed no significant difference in *q* (p = 0.11, MW U-test, *Figure 2F*). We found significant difference in $P_r$(at 1.5 mM $[Ca^{2+}]_o$: human: 0.33 ± 0.10, range: 0.18–0.48, n = 7; rat: 0.17 ± 0.13, range: 0.05–0.53, n = 12; p = 0.01, MW U-test).

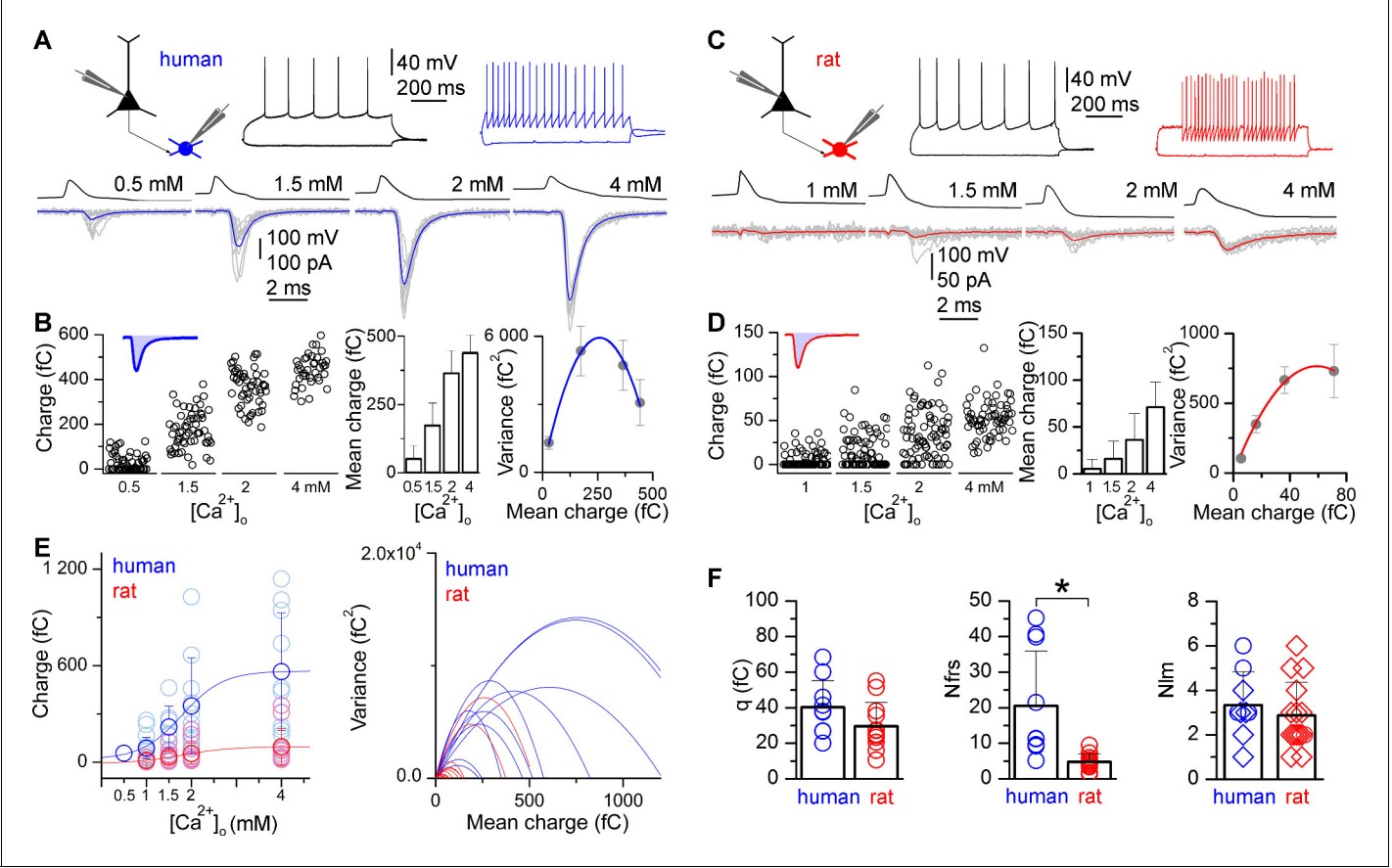

**Figure 2.** Higher number of functional release sites in human excitatory synapses is revealed by multiple probability fluctuation analysis. (**A**) Firing pattern of a human pyramidal cell (black) monosynaptically connected to a postsynaptic basket cell (blue). Bottom, Pyramidal cell action potential-evoked EPSCs in the basket cell recorded in 0.5, 1.5, 2 and 4 mM $[Ca^{2+}]_o$ (grey: 10 consecutive traces; blue: averages). (**B**) Left, Charge transferred by the unitary EPSCs recorded in different $[Ca^{2+}]_o$. Inset, The shaded blue area illustrates the charge of an EPSC. Middle, mean ± SD of the EPSC charge (in the same experiment). Right, parabolic fit to the variance versus mean of EPSC charge recorded in different $[Ca^{2+}]_o$, revealing an estimate for the number of functional release sites ($N_{frs}$ = 11.3) and the quantal size ($q$ = 45.9 fC). (**C–D**) A similar experiment is shown for a rat pyramidal cell (black) to basket cell (red) connection ($N_{frs}$ = 4.9; $q$ = 25.0 fC). (**E**) Left, Dependence of the EPSC charge in human (blue) and rat (red) on the $[Ca^{2+}]_o$ (1 mM: 87.3 ± 66.7 fC vs. 9.6 ± 12.7 fC; 1.5 mM: 218.7 ± 129.7 fC vs. 30.2 ± 37.6 fC; 2 mM: 349.7 ± 299.0 fC vs. 54.0 ± 66.1 fC; 4 mM: 563.2 ± 365.1 fC vs. 94.2 ± 113.8 fC). Right, Parabolic fits to the variance versus mean plots in human (blue) and rat (red). (**F**) Multiple probability fluctuation analysis of human and rat unitary EPSCs fails to show a significant difference in quantal size ($q$, human: 40.2 ± 14.9 fC, n = 10; rat: 29.6 ± 13.3 fC, n = 12; p = 0.11, MW U-test), but the $N_{frs}$ in human is substantially larger compared to rats (human: 20.5 ± 15.4, n = 10; rat: 4.7 ± 2.3, n = 12; p < 0.001, MW U-test). However, the number of LM detected contact sites ($N_{lm}$, right, human: 3.3 ± 1.5, n = 9; rat: 2.9 ± 1.5, n = 15) between presynaptic axons and postsynaptic dendrites is similar in human and rat (p = 0.36, MW U-test). Reconstructions with and without MPFA are shown by circles and diamonds, respectively. Data presented as mean ± SD.

Furthermore, our analysis revealed a 4.4-times larger $N_{frs}$ in human connections (20.5 ± 15.4, range 5.1–45.2) compared to rats (4.7 ± 2.2, range 1.3–9.5; p<0.001, MW U-test, *Figure 2F*).

## Structural properties of pyramidal cell to fast spiking interneuron connections in the human and rat cerebral cortex

A potential explanation for the differences in EPSC amplitudes and $N_{frs}$ between human and rat is a larger number of synaptic contacts connecting the PC axons to the postsynaptic IN dendrites in the human cortex. To assess the number of putative synaptic contacts, team members without access to the $N_{frs}$ searched for close appositions of presynaptic axon terminals and postsynaptic dendrites under light microscopy (LM). The presynaptic axon from the parent PC soma to the target IN dendrites were reconstructed in nine human (n = 2 PC to AAC and n = 7 PC to BC pairs, *Figure 1D*) and

fifteen rat (n = 15 PC to BC pairs, *Figure 1H*) pairs that partially overlapped with MPFA analyzed cell pairs. Comparison of the number of LM detected synaptic contacts ($N_{lm}$) between human (3.3 ± 1.5; range: 1–6) and rat (2.9 ± 1.5, range: 1–6) revealed no significant difference (p = 0.35, MW U-test, *Figure 2F*). Consequently, dividing the mean $N_{frs}$ with the mean $N_{lm}$ indicated that each pre-synaptic bouton contains on average 6.2 and 1.6 functional release sites in human and rat, respectively. This ratio was also larger for human connections when calculated from those pairs for which both MPFA and LM reconstructions could be performed (human: $N_{frs}/N_{lm}$ = 5.7 ± 5.1, range 1.6–13.3, n = 4; rat: $N_{frs}/N_{lm}$ = 2.6, n = 1). These results demonstrate that differences in the number of synaptic junctions between PCs and fast-spiking INs is not responsible for the observed 4.4-fold larger $N_{frs}$, therefore each presynaptic PC axon terminal must contain a larger $N_{frs}$ in human.

Finally we assessed potential ultrastructural correlates of the above described differences in the $N_{frs}$ in individual axon terminals. We performed three-dimensional reconstructions of presynaptic axon terminals from 20 nm thick serial EM sections (*Figure 3*). Biocytin-filled, *post hoc* recovered human (n = 3) and rat (n = 3) INs were chosen and axon terminals that established asymmetrical synaptic contacts on selected dendrites (see Supplementary Materials and methods) were fully reconstructed (*Figure 3A,C,H,J*). First we compared the size of presynaptic AZs and found that both human and rat active zones (AZs) have variable sizes (human: 0.02–0.26 $\mu m^2$; rat: 0.02–0.08 $\mu m^2$), but the human AZs were on average twice as large (*Figure 3P*). Each bouton contained only a single AZ in both species. When we counted docked vesicles in these AZs using the 20 nm thin serial section approach we found a significantly larger number in human (4.2 ± 2.2 / AZ) compared to rats (1.3 ± 0.8 / AZ; p<0.001, MW U-test; *Figure 3Q*), resulting in a twice as large docked vesicle density in human (*Figure 3R*). However, as the precise identification of docked vesicles is challenging in conventional EM even if very thin (20 nm) sections are used, we repeated these measurements using EM tomography; the best currently known method for this application (*Imig et al., 2014*; *Siksou et al., 2007*). Analysis of EM tomograms in 3 human (n = 33 synapses) and 3 rat (n = 31 synapses) samples revealed a similar two-fold difference in the mean docked vesicle density (*Figure 3R*). These results confirmed that reconstructions from very thin serial EM sections also provide a reliable measure of the number of docked vesicles. The large variability in the AZ area and number of docked vesicle prompted us to investigate their relationship and found a positive correlation between these parameters in both species (*Figure 3S*; human: ρ = 0.71; rat: ρ= 0.54, Spearman correlation). Moreover, the docked vesicle density and the size of the proximal vesicle pool normalized to the AZ area in axon terminals (n = 15 from both species, for definition see Methods) showed no significant correlation both in human (ρ = 0.15, P = 0.59, Spearman correlation), and rat samples (ρ = −0.09, p = 0.76; *Figure 3—figure supplement 1A*). Furthermore, we found significantly larger bouton volumes in humans (0.14 ± 0.09 $\mu m^3$, n = 20) compared to rats (0.05 ± 0.02 $\mu m^3$, n = 17; p<0.01, MW U-test; *Figure 3—figure supplement 1B*).

## Discussion

Paired recordings, MPFA and *post hoc* anatomical reconstructions revealed that presynaptic AZs contain on average 6 functional release sites in human, but only 1.6 in rats. High resolution EM analysis identified corresponding species-specific differences in the AZ area and the number of docked vesicles. Our data allowed us to provide the first estimate of the size of Katz's functional release site (*del Castillo and Katz, 1954*) of cortical synapses; approximately 0.012 $\mu m^2$ AZ membrane area in human and 0.025 $\mu m^2$ in rats. Thus, the space that harbors a functional release site or a docking site seems to be substantially smaller in human AZs. This raises an interesting question: why the molecular machinery necessary for the assembly of a functional release site needs less space in human? Answering this question requires quantitative proteomic analysis of these AZs at nanometer resolutions. Furthermore, it will be also interesting to see whether this species-specific difference is valid for all central synapses or it is a unique feature of the cortical microcircuit.

Our data demonstrating that an AZ in rat PC axons contains on average a single functional release site is consistent with the 'one site-one vesicle' hypothesis (*Korn et al., 1982*) and with data previously published for rodent cortical and hippocampal excitatory synapses (*Biro et al., 2005*; *Gulyas et al., 1993*; *Silver et al., 2003*). In contrast, the same type of axon terminals in human contain on average 6 functional release sites, demonstrating that here multivesicular release (MVR; (*Rudolph et al., 2015*) is the main mode of operation. However, it is important to note that our data

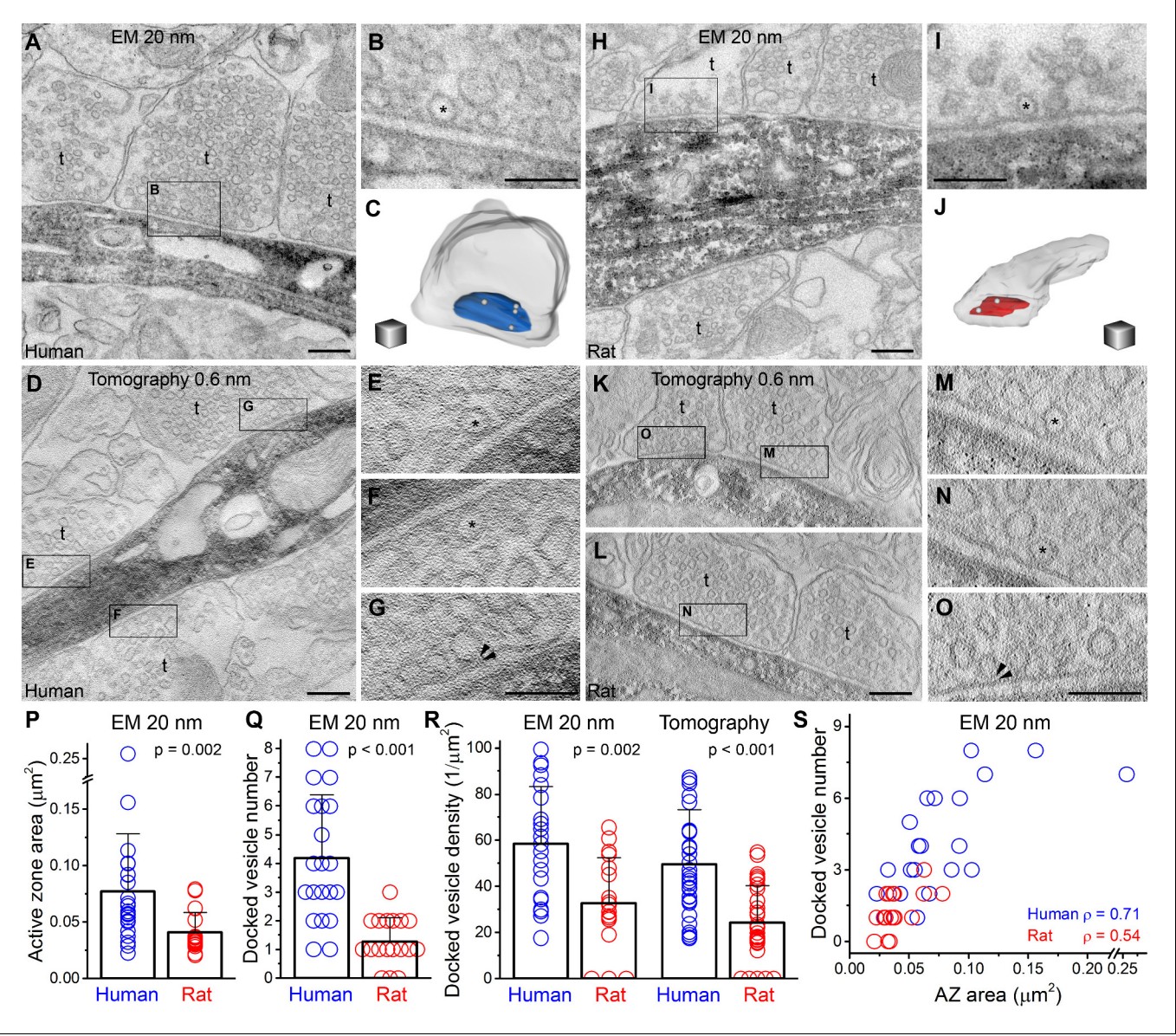

**Figure 3.** Active zones (AZs) of excitatory synapses are twice as large and harbor 4 times more docked vesicles in human compared to rats. (A, H) Electron microscopic images of 20 nm thick sections show axon terminals forming asymmetrical synapses on a human (A) and a rat (H) FS cell dendrite (intracellularly filled interneuron visualized with peroxidase: dark precipitate). (B, I) Higher magnification views of the boxed areas in A and H, showing docked vesicles (*) at the AZs. (C, J) 3D reconstructions of the terminals shown in B and I. The human terminal (c, grey semitransparent contour) contains an AZ (blue) of 0.09 $\mu m^2$ with 4 docked vesicles (white spheres), whereas the rat AZ (red in J) is 0.04 $\mu m^2$ and has 2 docked vesicles. (D, K–L) Electron tomographic subvolumes (0.6 nm thick) of human (D) and rat (K–L) axon terminals (t) that establish asymmetrical synaptic contacts on FS cell dendrites (dark precipitate). (E–G, M–O) Boxed areas from D and K-L show docked vesicles (*). Panels G and O show membrane proximal vesicles with distances smaller than 5 nm (distance between arrowheads). (P) The area of AZs, determined from 3D reconstructions from 20 nm serial sections, is twice as large in humans than in rats (0.077 ± 0.051 $\mu m^2$, n = 22, from 3 separate human samples; rat: 0.041 ± 0.017 $\mu m^2$, n = 19 from 3 animals; p = 0.002, MW U-test). (Q) The number of the docked vesicles, identified in fully reconstructed AZs from 20 nm serial sections, is 4-times larger in human compared to rats (human: 4.2 ± 2.2, n = 21; rat: 1.3 ± 0.8, n = 18, p<0.001, MW U-test). (R) The density of docked vesicles, measured either in 20 nm reconstructions (human: 58.5 ± 24.6 / $\mu m^2$, n = 21, rat: 32.5 ± 19.9 / $\mu m^2$, n = 18, p < 0.001, unpaired t-test) or in EM tomographic volumes (human: 49.5 ± 23.8 / $\mu m^2$, n = 33, rat: 24.3 ± 16.0 / $\mu m^2$, n = 31, P < 0.001, MW U-test), is significantly different between the two species. (S) The number of the docked vesicles shows a positive correlation with the AZ area (Spearman correlation). Scale bars: A, D, H, K, L: 200 nm, B, E-G, I, M-O: 100 nm, C, J: side of the cubes: 200 nm. Data presented as mean ± SD.

The following figure supplement is available for figure 3:

**Figure supplement 1.** Ratio of docked vesicles in the presynaptic vesicle pool and bouton volumes in human and rat axon terminals.

clearly demonstrate large variability in the AZ area in both species and that the number of docked vesicles and docking sites correlates with the AZ area. Thus, MVR is not a human-specific feature of cortical synapses, but it is expected to occur at large AZs of this connection in rats, as it has been shown to take place in many other rat glutamatergic synapses (*Rudolph et al., 2015*). Comparison of the $N_{frs}$ and the number of docked vesicles allowed the calculation of the average docking site occupancy that is 0.8 in rats and ~0.7 in human synapses, similar to that calculated for the Calyx of Held (*Neher, 2010*), and somewhat higher than that found in cerebellar IN synapses (*Pulido et al., 2015*). Similarly large docking site occupancy and vesicle $P_r$ between human and rodents predict similar short-term plasticity of transmission, a key feature of synapses of dynamic neuronal networks, with an additional tuning range for human connection strength due to higher $N_{frs}$.

Our results provide an in depth understanding of the mechanisms underlying the powerful, unitary EPSP-evoked suprathreshold series of events in the human neocortex (*Molnar et al., 2008*), which might be an essential cellular component underlying the generation of Hebbian-like cell assemblies. Even if upscaling of release components simply follows larger neurons and synapses in the human microcircuit, it is important to note that scalability of multivesicular release in human synapses as opposed to all-or-none single vesicular rat synapses provides additional functional variability, which might subserve evolutionary optimization. The strong connections linking pyramidal cells and a select group of postsynaptic neurons in the same area of the neocortex could be strategically positioned for the emergence and maintenance of key cortical hubs characteristic of the human cerebral cortex (*Haggmann et al. 2008*). Long-range connections of PCs with strong local connections might also target distant INs with similarly powerful unitary EPSPs speculatively providing structural and functional basis for the formation of small world networks suggested to be important in linking connections to cognition (*Park and Friston, 2013*).

## Materials and methods

### Electrophysiological recordings

All procedures were performed according to the Declaration of Helsinki with the approval of the University of Szeged Ethical Committee. Human slices were derived from material that had to be removed to gain access for the surgical treatment of deep-brain tumors from the left and right frontal (n = 22), temporal (n = 14), and parietal (n = 3) regions with written informed consent of female (n = 15, aged 53 ± 13 years) and male (n = 11, aged 43 ± 24 years) patients prior to surgery. We found no significant differences between EPSC charge values between experiments performed in slices prepared from frontal and temporal lobes (425.8 ± 426.7 vs. 760.6 ± 761.2, p = 0.1, MW U-test) and between morphologically identified basket and axo-axonic cell groups (567.5 ± 451.6 vs. 364.9 ± 193.3, p = 0.36, MW U-test). Furthermore, we found no significant differences in EPSC parameters measured in female and male patients (charge, female: 602.49 ± 480.81 fC, male: 499.27 ± 708.32 fC,p =0.13; rise time, female: 0.46 ± 0.20 ms, male: 0.42 ± 0.13 ms, p = 0.71; decay time, female:1.32 ± 0.56 ms, male:1.63 ± 1.04, p = 0.65, latency female:1.08 ± 0.46 ms, male: 0.99 ± 0.37 ms, p = 0.7, MW U-test). In addition, we found no correlation between the age of patients and the charge (ρ = 0.05, p = 0.76), rise time (ρ = 0, p = 0.99), latency (ρ = 0.16, p = 0.32) and decay (ρ = 0.01, P = 0.94) of the measured EPSCs (Spearman correlation, *Figure 1—figure supplement 1*). Anesthesia was induced with intravenous midazolam and fentanyl (0.03 mg/kg, 1–2 µg/kg, respectively). A bolus dose of propofol (1–2 mg/kg) was administered intravenously. To facilitate endotracheal intubation, the patient received 0.5 mg/kg rocuronium. After 120 s, the trachea was intubated and the patient was ventilated with a mixture of $O_2$ and $N_2O$ at a ratio of 1:2. Anaesthesia was maintained with sevoflurane at monitored anaesthesia care (MAC) volume of 1.2–1.5. After surgical removing blocks of tissue were immersed immediately in ice-cold solution containing (in mM) 130 NaCl, 3.5 KCl, 1 NaH$_2$PO$_4$, 24 NaHCO$_3$, 1 CaCl$_2$, 3 MgSO$_4$, 10 d(+)-glucose, saturated with 95% $O_2$ and 5% $CO_2$. Slices were cut perpendicular to cortical layers at a thickness of 350 µm with a vibrating blade microtome (Microm HM 650 V) and were incubated at room temperature for 1 hr in the same solution. The solution used during recordings was similar to the slicing solution, but it contained 3 mM CaCl$_2$ and 1.5 mM MgSO$_4$. Coronal slices (350 µm) were prepared from the somatosensory cortex of male Wistar rats (P18–P28, n = 9, RRID:RGD_2312511) and medial prefrontal cortex of adult male Wistar rats (P53–65, n = 17) as described previously (*Molnar et al., 2014*). We

found no significant differences in EPSC charge values between the two rodent groups (81.6 ± 59.3 vs. 72.7 ± 60.0, p = 0.91, MW U-test). We calculated the relative age of rats involved in the study referenced to human age taking into account that 3.3 rat days equals one human year around sexual maturity (P38 in rats and 12 years in human, *Sengupta, 2013*). Thus, the age range of rats used in our experiments corresponds to approximately 6–20 years in humans. Recordings were performed at 36°C. Cells were visualized through water-immersion 40× objective using infrared differential interference contrast videomicroscopy at depths 60–130 µm from the surface of the slice. Micropipettes (3.5–5 MΩ) were filled with two types of low $[Cl]_i$ intracellular solution for whole-cell patch-clamp recording: (in mM): 126 K-gluconate, 4 KCl, 4 ATP-Mg, 0.3 GTP-Na₂, 10 HEPES, 10 phosphocreatine, and 8 biocytin (pH 7.20; 300 mOsm), of the postsynaptic INs and 116 K-gluconate, 10 L-glutamic acid, 4 KCl, 4 ATP-Mg, 0.3 GTP-Na₂, 10 HEPES, 10 phosphocreatine, and 8 biocytin (pH 7.20; 300 mOsm) for presynaptic PCs. Membrane currents were recorded with a dual-channel amplifier an EPC 10 quadro amplifier (HEKA) filtered at 7 or 8 kHz (Bessel filter), digitized at 50–100 kHz, and analyzed with PULSE software. Series resistance (Rs) and whole-cell capacitance was checked continuously in the postsynaptic IN. The mean Rs for the first analyzed epochs during the recordings was 19.7 ± 6.9 MΩ and after 66–73% compensation it was 6.0 ± 2.2 MΩ. The experiment was discarded if the compensated Rs change reached 20% during the recording. We performed multiple probability fluctuation analysis (MPFA) by altering $[Ca^{2+}]_o$ and $[Mg^{2+}]_o$ in three to five different conditions (*Biro et al., 2006*; *Silver, 2003*; *Silver et al., 1998*) from lower to higher $Ca^{2+}$ concentrations adding the following substances to the recording ACSF: 20 µM DL-2-Amino-5-phosphonopentanoic acid (AP-5) (Tocris), 10 µM 1-(2,4-Dichlorophenyl)-5-(4-iodophenyl)-4-methyl-N-1-piperidinyl-1H-pyrazole-3-carboxamide (AM251) (Sigma-Aldrich, St. Louis, MO), 0.5 mM γ-D-glutamylglycine (γDGG) (Abcam). PCs were stimulated with brief (3–15 ms) current pulses (+ 850 pA) delivered at 5 s intervals, the postsynaptic membrane potential was held at −70 mV. The stability of peak amplitudes or charge values in time throughout an epoch (73 and 78 events on average in each epoch in human and rat, respectively) was determined by running average and by checking fulfillment of slope <0.05 relation of the fitted a regression line to the scatter plot of EPSC charge vs. time. Elapsed time from tissue removal until the start of MPFA experiments was 9.8 ± 4.1 and 5.3 ± 1.8 hr in the human and rat, respectively. In order to exclude unwanted decrease of EPSC amplitudes during long recordings, we performed control experiments on pyramidal cell to basket cell pairs (n = 5). During these experiments we used the same configuration to assess possible rundown effects. We made at least 50 min-long recordings (up to 90 min) with inter-event intervals of 5 s. We found no significant decrease in the amplitude or charge between epochs selected from the first and last part of recordings (50 traces, charge: p≥0.41; amplitude: p≥0.21, unpaired t-test).

## Data analysis

Data were analyzed with Fitmaster (HEKA), Origin 7.5 (OriginLab), and Igor Pro (Wavemetrics). Detection and analysis of EPSCs was performed with NeuroMatic software (http://www.neuromatic.thinkrandom.com). Detection threshold was set to 2 to 4 times the standard deviation of the noise, onset time limit was set to 2 ms, which defines the maximum interval from the baseline to the deflection reaches the threshold. Peak time limit was set to 3 ms. All detected events were inspected by eye. In order to minimize variance originating from temporal fluctuations, we calculated variances using the pairwise method (see [*Biro et al., 2006*] and [*Scheuss et al., 2002*], Eq(2)). Then to obtain q and $N_{frs}$ values we plotted the pairwise variance versus mean and fitted with parabola. Data points were weighted by the theoretical variance of variance (*Silver, 2003*), Eqs. 22–25.

## Statistics

Data are presented as mean ± standard deviation (SD), statistical tests are defined for each paradigm; differences were accepted as significant if p<0.05.

## Histology and reconstruction

After electrophysiological recordings slices were fixed in a fixative containing 4% paraformaldehyde, 15% picric acid and 1.25% glutaraldehyde in 0.1 M phosphate buffer (PB; pH = 7.4) at 4°C for at least 12 hr. After several washes in 0.1 M PB, slices were cryoprotected in 10% then 20% sucrose solution in 0.1 M PB. Slices were frozen in liquid nitrogen then thawed in PB, embedded in 10%

gelatin and further sectioned into slices of 60 μm in thickness. Sections were incubated in a solution of conjugated avidin-biotin horseradish peroxidase (ABC; 1:100; Vector Labs) in Tris-buffered saline (TBS, pH = 7.4) at 4°C overnight. The enzyme reaction was revealed by 3'3-diaminobenzidine tetra-hydrochloride (0.05% ) as chromogen and 0.01% $H_2O_2$ as oxidant. Sections were post fixed with 1% $OsO_4$ in 0.1 M PB. After several washes in distilled water, sections were stained in 1% uranyl acetate, dehydrated in ascending series of ethanol. Sections were infiltrated with epoxy resin (Durcupan) overnight and embedded on glass slices. Three dimensional light microscopic reconstructions were carried out using Neurolucida system with 100x objective.

### Electron microscopy

Visualization of biocytin and correlated light and electron microscopy (LM and EM) were performed as described previously (*Olah et al., 2009*; *Szabadics et al., 2006*). Dendritic segments (distance from the soma ~50 μm in rat and ~150 μm in human) of biocytin filled basket cells (identified based on distinctive electrophysiological properties and LM investigation of the axonal arbor) were re-embedded and re-sectioned at 20 nm thickness. Digital images of serial EM sections were taken at 64 000 magnification with a FEI/Philips CM10 electron microscope equipped with a MegaView G2 camera. Dendrites and the presynaptic boutons were 3D-reconstructed using the Reconstruct software (http://synapses.clm.utexas.edu/). The areas of active zones were measured at perpendicularly cut synapses, where the rigid apposition of the pre- and postsynaptic membranes was clearly visible, and the docked vesicles were identified as described previously (*Holderith et al., 2012*). Active zones were identified by the parallel rigid membrane appositions where the synaptic cleft widened (because the PSD is masked by the dark DAB precipitate). Potential inhibitory synapses with flattened vesicles were discarded from the analysis. Bouton volumes of a subpopulation of boutons were measured from 20 nm serial reconstructions where the series contained the whole terminal (n = 9 human boutons, n = 8 rat boutons). We calculated the volume of the same subset of boutons from the area of the largest cross-section (assuming that boutons are spherical) and found a tight correlation ($\rho$ = 0.9019, Spearman correlation) with the measured volume. Based on this correlation, we calculated the volume from the largest cross-section of those terminals where the AZ was measured, but the series did not contain the whole terminal.

### EM tomography

200 nm thick sections were cut and collected onto copper slot grids. Fiducial markers (Protein-A conjugated to 10 nm gold particles, Cytodiagnostics - Absource Diagnostics GmbH, Munich, Germany) for tomographic reconstruction were introduced at both sides of the grids as described by Imig et al. (*Imig et al., 2014*). Single-axis tilt series of perpendicularly oriented synapses were acquired in FEI Tecnai G2 Spirit BioTWIN transmission EM operating at 120 kV and equipped with an Eagle 4K HS digital camera (FEI Europe Nanoport, Eindhoven, The Netherlands). Tilt series were recorded between ± 65° (with 2° increments between ± 45° and with 1° increments between 45–65°) at a magnification of 30,000 using FEI Xplore3D for automated tilt series acquisition. Tomographic volumes were reconstructed using the IMOD package (*Imig et al., 2014*; *Kremer et al., 1996*) and exported as z-stacks for analysis. Active zone area and vesicle distance from the presynaptic membrane were measured with Reconstruct software. A vesicle was considered to be docked if the outer part of the lipid bilayer was in direct contact with the inner part of the presynaptic membrane bilayer. A vesicle was considered to be pre-docked if the distance between these lipid bilayers did not exceed 5 nm, and the membrane proximal vesicles are defined as the docked and pre-docked vesicles. For vesicle pool measurements, synaptic vesicles within 100 nm of the AZ were quantified from tomographic subvolumes and normalized to the AZ area according to *Imig et al.,(2014)* in human and rat samples (n = 15 each).

## Acknowledgements

GT is supported by the ERC INTERIMPACT project, the Hungarian Brain Research Program and the Hungarian Academy of Sciences. ZN is the recipient of a Hungarian Academy of Sciences Momentum Grant (Lendület, LP2012-29) and a European Research Council Advanced Grant (293681). NH is funded by Janos Bolyai Scholarships of the Hungarian Academy of Sciences. The financial support from these agencies is gratefully acknowledged. We would like to thank Nelli Tóth, Éva Tóth and

Dóra Rónaszéki for their excellent technical assistance, Csaba Cserép for his vital help in EM tomography and Drs Karri Lamsa, Erwin Neher and Angus Silver for their comments on the MS.

## Additional information

### Funding

| Funder | Grant reference number | Author |
| --- | --- | --- |
| European Research Council | INTERIMPACT | Gábor Tamás |
| European Research Council | 293681 | Zoltan Nusser |
| Magyar Tudományos Akadémia | MTA-SZTE Agykergi Neuronhalozatok Kutatocsoport | Gábor Tamás |
| Magyar Tudományos Akadémia | Lendület, LP2012-29 | Zoltan Nusser |
| Magyar Tudományos Akadémia | Janos Bolyai Scholarship | Noémi Holderith |
| Nemzeti Kutatási és Technológiai Hivatal | VKSZ_14-1-2015-0155 | Gábor Tamás |
| Nemzeti Agykutatasi Program | NAP-A | Gábor Molnár |

The funders had no role in study design, data collection and interpretation, or the decision to submit the work for publication.

### Author contributions

GM, MR, JB, NH, PB, Acquisition of data, Analysis and interpretation of data; ZN, GT, Conception and design, Analysis and interpretation of data, Drafting or revising the article

### Author ORCIDs

Noémi Holderith, http://orcid.org/0000-0002-0024-3980
Zoltan Nusser, http://orcid.org/0000-0001-7004-4111

### Ethics

Human subjects: All procedures were performed according to the Declaration of Helsinki with the approval of the University of Szeged Ethical Committee. Informed consent, and consent to publish, was obtained from patients. The permit number for our human experiments is 75/2004 issued by the Human Investigation Review Board of the University of Szeged.

Animal experimentation: All experimental protocols and procedures were performed according to the European Communities Council Directives of 1986 (86/609/EEC) and 2003 (2003/65/CE) for animal research and were approved by the Ethics Committee of the University of Szeged.

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
