## [Decision Letter]

Thank you for submitting your article "Functional differences between human and rat neocortical excitatory synapses" for consideration by *eLife*. Your article has been favorably evaluated by Eve Marder as the Senior editor and two reviewers: Marlene Bartos (Reviewer #1), who is a member of our Board of Reviewing Editors and Imre Vida (Reviewer #2).

The reviewers have discussed the reviews with one another and the Reviewing Editor has drafted this decision to help you prepare a revised submission.

Summary:

The study by Molnar et al. addresses functional and ultrastructural differences between excitatory synapses in the rat and the human neocortex focusing on a well-defined connection between pyramidal cells and fast-spiking perisomatic inhibitory basket or axo-axonic cells by applying in vitro paired recordings combined with light and electron microscopic analysis. The main finding of the study is that at human synapses EPSCs have larger mean amplitudes mediated by a higher number of release sites in comparison to those in the rat. This functional difference correlates well with the divergent number of docked vesicles at presynaptic active zones in the two species. The work was judged by both reviewers as excellent. The results are of high quality, well-presented and the conclusions are supported by the data. The results provide important insights into how synapses operate in the human cortex and what differences may underlie the superior performance of human circuits despite the general similarities in terms of anatomical and physiological properties of neocortical synapses, cells and networks across various mammalian species.

The criticism formulated by the reviewers addresses very specific aspects of the manuscript and are therefore listed below:

Essential revisions:

1) Did the authors observe any differences in the dynamic properties (short term or long term dynamics) of excitatory signals during paired recordings when trains of presynaptic spikes were evoked? Since synaptic plasticity has been identified at glutamatergic synapses targeting basket cells in the rodent cortex, it would be of high interest to see whether this form of plasticity also exist at principal cell to basket cell synapses in human tissue. If the data are available then they should be added to the study.

2) The reviewer was wondering how the age range in humans in this study (16-75 y) may be compared to the age range of rats (P18-65). Please provide available age relationships in the Methods section.

3) It would be interesting to the readership to include additional quantitative morphological data such as axonal and dendritic length from human basket and axo-axonic cells. The reconstructions in Figure 1 illustrates that these data should be available. If so, please include them to the Results section.

4) Please move the extensive data set from Figure legend 1I to the Results section.

5) The authors should provide some explanations in the Introduction on why they have chosen specifically this synapse to address their questions.

6) Please answer the following questions and include the answers to the manuscript: From which layer were the recordings made? Why have been recordings performed in the presence of NMDA and a cannabinoid receptor antagonist? How is the number of docked vesicles related to the size of the vesicle pool per axon terminal? How do the size of the axon terminals (diameter and/or volume) differ in the human vs. rat brain? Provide information about the distribution of the axon terminal size.

7) The Methods section requires criteria for the identification of the active zone. Also, the identification of asymmetrical vs. symmetrical synapses on biocytin stained non-spiny/sparsely spiny dendrites is not trivial – please provide the criteria used.

8) One reviewer suggested to extend the Discussion by addressing the question whether the observed differences may be the result of "evolutionary optimization" to achieve a more intelligent brain or simply a byproduct of nonhomogeneous scaling of the nervous system (i.e. larger brain, larger neurons and synapses but unchanged release components will inevitably lead to increased number of structural release elements). Although this is beyond the scope of the current study, it would be interesting to address this in the Discussion.

---

## [Author Response]

*The criticism formulated by the reviewers addresses very specific aspects of the manuscript and are therefore listed below:*

Essential revisions:

*1) Did the authors observe any differences in the dynamic properties (short term or long term dynamics) of excitatory signals during paired recordings when trains of presynaptic spikes were evoked? Since synaptic plasticity has been identified at glutamatergic synapses targeting basket cells in the rodent cortex, it would be of high interest to see whether this form of plasticity also exist at principal cell to basket cell synapses in human tissue. If the data are available then they should be added to the study.*

We agree that plasticity of pyramidal to basket cell synapses is of special interest and actually dedicated a separate study to this topic (under review). In case you need further information, please contact us.

*2) The reviewer was wondering how the age range in humans in this study (16-75 y) may be compared to the age range of rats (P18-65). Please provide available age relationships in the Methods section.*

We calculated the relative age of rats involved in the study referenced to human age taking into account that 3.3 rat days equals one human year around sexual maturity (P38 in rats and 12 years in human, Sengupta, 2013). Thus, the age range of rats used in our experiments corresponds to approximately 6-20 years in humans. We added this sentence to the Methods section.

Sengupta, P., 2013. The Laboratory Rat: Relating Its Age With Human’s. Int. J. Prev. Med. 4, 624–30.

*3) It would be interesting to the readership to include additional quantitative morphological data such as axonal and dendritic length from human basket and axo-axonic cells. The reconstructions in Figure 1 illustrates that these data should be available. If so, please include them to the Results section.*

Indeed we have a number of reconstructions representing the full extent of dendrites (basket cells, n = 9, 2912 ± 1076 µm; axo-axonic cells, n = 5, 2897 ± 1546 µm) and axons (basket cells, n = 7, 27489 ± 14080 µm; axo-axonic cells, n = 5, 20732 ± 3332 µm) of human neurons. Now, we have included the new data to the Results section as requested. Dendritic and axonal lengths were not significantly different between basket and axo-axonic cells (P=1 and P=0.41 respectively, Mann-Whitney U test).

*4) Please move the extensive data set from Figure legend 1I to the Results section.*

Done.

*5) The authors should provide some explanations in the Introduction on why they have chosen specifically this synapse to address their questions.*

We have changed the Introduction as requested.

*6) Please answer the following questions and include the answers to the manuscript: From which layer were the recordings made? Why have been recordings performed in the presence of NMDA and a cannabinoid receptor antagonist? How is the number of docked vesicles related to the size of the vesicle pool per axon terminal? How do the size of the axon terminals (diameter and/or volume) differ in the human vs. rat brain? Provide information about the distribution of the axon terminal size.*

Recordings were made from layer 2/3, we modified the first sentence of the Results section.

Recordings were performed in the presence of an NMDA and a cannabinoid receptor antagonist (20 μM D-AP-5 and 10 μM AM251, respectively) to exclude NMDA channel openings which might reduce variances and induce long-term plasticity (Silver, 2003; Saviane and Silver, 2007) and to exclude undesirable retrograde modification of glutamatergic transmission (Lee et al. 2010, Péterfi et al. 2012) during MPFA. We changed the Results accordingly.

We extended our analysis to reveal potential relationships between the docked vesicle density and the size of the proximal vesicle pool normalized to the AZ area in axon terminals in our human and rat samples (n = 15 axon terminals from each species) and found no correlation (human: Spearman’s ρ = 0.15, P = 0.59; rat: Spearman’s ρ = -0.09, P=0.76) (Figure 3—figure supplement 1).

By measuring bouton areas we calculated the volume of axon terminals (see revised Methods) and found significantly higher volumes in human (0.14 ± 0.09 µm^3^, n=20) compared to the rat samples (0.05 ± 0.02 µm^3^, n=17). The distribution of axon terminal size is presented on the new Figure 3—figure supplement 1)

*7) The Methods section requires criteria for the identification of the active zone. Also, the identification of asymmetrical vs. symmetrical synapses on biocytin stained non-spiny/sparsely spiny dendrites is not trivial – please provide the criteria used.*

Active zones were identified by the parallel rigid membrane appositions where the synaptic cleft widened (as PSD is not visible in the dark precipitate of the DAB). Potential inhibitory synapses with flattened vesicles were discarded from the analysis. We changed the Materials and methods section accordingly.

*8) One reviewer suggested to extend the Discussion by addressing the question whether the observed differences may be the result of "evolutionary optimization" to achieve a more intelligent brain or simply a byproduct of nonhomogeneous scaling of the nervous system (i.e. larger brain, larger neurons and synapses but unchanged release components will inevitably lead to increased number of structural release elements). Although this is beyond the scope of the current study, it would be interesting to address this in the Discussion.*

We believe (without causal evidence presented in the manuscript) that a multitude of postsynaptic responses and potential network effects generated by multivesicular release could lead to advantages in an evolutionary process. We modified the last paragraph of the Discussion accordingly:

“Our results provide an in depth understanding of the mechanisms underlying the powerful, unitary EPSP-evoked suprathreshold series of events in the human neocortex (Molnar et al., 2008), which might be an essential cellular component underlying the generation of Hebbian-like cell assemblies. […] Long-range connections of PCs with strong local connections might also target distant INs with similarly powerful unitary EPSPs speculatively providing structural and functional basis for the formation of small world networks suggested to be important in linking connections to cognition (Park and Friston, 2013).”